# Orchestrated Response of Intracellular Zwitterionic Metabolites in Stress Adaptation of the Halophilic Heterotrophic Bacterium *Pelagibaca bermudensis*

**DOI:** 10.3390/md20110727

**Published:** 2022-11-19

**Authors:** Muhaiminatul Azizah, Georg Pohnert

**Affiliations:** 1Bioorganic Analytics, Institute for Inorganic and Analytical Chemistry, Friedrich Schiller University, Lessingstrasse 8, D-07743 Jena, Germany; 2MPG Fellow Group, Max Planck Institute for Chemical Ecology, Hans-Knöll-Straße 8, D-07745 Jena, Germany

**Keywords:** *Pelagibaca bermudensis*, *Tetraselmis striata*, osmoadaption, thermoadaptation, DMSOP, DMSP, gonyol, zwitterionic metabolites, high-resolution mass spectrometry, ZIC-HILIC, cysteinolic acid

## Abstract

Osmolytes are naturally occurring organic compounds that protect cells against various forms of stress. Highly polar, zwitterionic osmolytes are often used by marine algae and bacteria to counteract salinity or temperature stress. We investigated the effect of several stress conditions including different salinities, temperatures, and exposure to organic metabolites released by the alga *Tetraselmis striata* on the halophilic heterotrophic bacterium *Pelagibaca bermudensis*. Using ultra-high-performance liquid chromatography (UHPLC) on a ZIC-HILIC column and high-resolution electrospray ionization mass spectrometry, we simultaneously detected and quantified the eleven highly polar compounds dimethylsulfoxonium propionate (DMSOP), dimethylsulfoniopropionate (DMSP), gonyol, cysteinolic acid, ectoine, glycine betaine (GBT), carnitine, sarcosine, choline, proline, and 4-hydroxyproline. All compounds are newly described in *P. bermudensis* and potentially involved in physiological functions essential for bacterial survival under variable environmental conditions. We report that adaptation to various forms of stress is accomplished by adjusting the pattern and amount of the zwitterionic metabolites.

## 1. Introduction

Environmental stress, such as changes in salinity, temperature, and imbalance of nutrients, leads to stress in marine bacteria. Adaptation to the changing environment is essential for the survival of these cells. As a response to environmental changes, they accumulate, produce, or uptake osmolytes. Osmolytes are low-molecular-weight metabolites that help maintain the cells’ biological functions even under severe stress [1,2,3,4]. Organic osmolytes are classified into the groups of (i) sugars, (ii) amino acids, and (iii) zwitterionic metabolites (sulfur-containing zwitterionic metabolites and nitrogen-containing zwitterionic metabolites) [5,6,7]. Dimethylsulfoniopropionate (DMSP), a sulfur-containing zwitterionic metabolite, is the main osmolyte found in many phytoplankton and marine bacteria with osmoregulatory and thermoregulatory functions [3,8]. Besides DMSP, many other highly polar metabolites that can serve as compatible solutes have been discovered in marine phytoplankton, such as the nitrogen-containing zwitterionic metabolite glycine betaine (GBT) [3]. The analysis of zwitterionic metabolites is challenging due to the limitation of analytical techniques for determining highly polar compounds [3]. The introduction of a ZIC-HILIC separation protocol combined with high-resolution mass spectrometry analysis has facilitated the systematic investigation of zwitterionic metabolites and other polar metabolites in marine organisms [3,9]. The recent discovery and identification of dimethylsulfoxonium propionate (DMSOP), a DMSP derivative biosynthetically produced by the marine heterotrophic bacterium *Pelagibaca bermudensis*, was, for example, facilitated by this analytical approach [10,11]. DMSOP is the latest addition of sulfur-containing zwitterionic metabolites with an unknown fate in the environment [11]. In addition, using this analytical approach, we also successfully identified cysteinolic acid produced by marine phytoplankton during osmotic stress [3]. While the invention of zwitterionic metabolites in marine bacteria is growing, to the best of our knowledge, the physiology and regulation of these metabolites in bacteria are poorly understood.

We selected the marine halophilic heterotrophic bacterium *P. bermudensis* as a study organism to elucidate the response of zwitterionic metabolites under stress. The marine halophilic bacterium *P. bermudensis* belongs to the Alphaproteobacteria class, a predominant and widely distributed bacterial class in the ocean [12,13]. *P. bermudensis* is found under diverse environmental conditions and thus needs efficient adaptation strategies. *P. bermudensis* HTCC2601 has genes encoding for the production of the highly polar DMSP and is also a DMSOP producer [10,11]. To our knowledge, the physiological regulation of the zwitterionic metabolites in *P. bermudensis* and most other marine bacteria remains unexplored. Furthermore, some osmolytes from algae have not been previously investigated regarding their role in marine heterotrophic bacterial osmotic stress response [14]. Since *P. bermudensis* is found to be associated with algae, we decided to investigate here the cellular content of zwitterionic metabolites under stress and upon interaction with the microalgae, *Tetraselmis striata* [15,16]. Adaptation to various forms of stress is accomplished by a collaborative adjustment of several different zwitterionic metabolites [17].

## 2. Results and Discussion

Using ultra-high-pressure liquid chromatography (UHPLC) separation on a ZIC-HILIC column and high-resolution (ESI-orbitrap) mass spectrometry detection, we simultaneously monitored eleven highly polar compounds potentially involved during stress adaptation in the halophilic heterotrophic bacterium, *P. bermudensis*. We detected known sulfur-containing zwitterionic metabolites (DMSOP, DMSP, gonyol, and cysteinolic acid) and nitrogen-containing zwitterionic metabolites (ectoine, glycine betaine, carnitine, sarcosine, choline, proline, and 4-hydroxyproline). In this study, dimethylsulfonioacetate (DMSA), one of the major osmoprotectants from the gammaproteobacterium *E. coli,* was not detected in any treatment of *P. bermudensis* [18]. Compounds were identified by comparison of their retention time, mass spectra, and MS/MS data with commercial (GBT, l-ectoine, l-carnitine, choline, sarcosine, l-proline, and l-4-hydroxyproline) and synthetic reference standards (DMSOP, DMSP, gonyol, and cysteinolic acid). The molecular ion traces of DMSOP (*m*/*z* 151.04231 for [M + H]^+^), DMSP (*m*/*z* 135.04753 for [M + H]^+^), gonyol (*m*/*z* 179.07376 for [M + H]^+^), cysteinolic acid (*m*/*z* 156.03239 for [M + H]^+^), ectoine (*m*/*z* 143.08162 for [M + H]^+^), sarcosine (*m*/*z* 90.05499 for [M + H]^+^), carnitine (*m*/*z* 162.11246 for [M + H]^+^), glycine betaine (*m*/*z* 118.08638 for [M + H]^+^), choline (*m*/*z* 104.10696 for [M + H]^+^), proline (*m*/*z* 116.07067 for [M + H]^+^), and 4-hydroxyproline (*m*/*z* 132.06567 for [M + H]^+^) were used for quantification. The chromatographic profile of zwitterionic metabolites from *P. bermudensis,* including Total Ion Count (TIC) and ion traces of zwitterionic metabolites, is shown in Appendix A. MS/MS data of each osmolyte compared to its reference standard are documented in spectral mirror charts (Appendix A). All MS/MS data are deposited and publicly available online in the GNPS spectral libraries (https://gnps.ucsd.edu/) (Accessed on 13 August 2022). All compounds were quantified by adding synthetic labeled internal standards.

To monitor the growth of all replicates and to determine the late exponential phase for collecting samples, OD_600_ was measured (Appendix A). Growth had a salinity-dependent optimum at 0.33 M. At elevated salinity, growth substantially decreased throughout the culture period (Appendix A), and at low salinity, OD_600_ dropped below that of the control after ca. 40 h of culturing (Appendix A). Growth at 28 °C and 38 °C was similar, while at 18 °C, growth was substantially reduced (Appendix A). The growth of *P. bermudensis* was not affected by exposure to *T. striata* extracts even under elevated concentrations (Appendix A).

### 2.1. Response to Osmotic Stress

The halophilic bacterium *P. bermudensis* encounters a wide range of NaCl concentrations in marine environments [13]. To mimic these conditions, the salinity of the medium was adjusted by modifying the NaCl content. The concentration of other nutrients in the media remained the same. The selected NaCl concentration range of media (0.03 M equal to 0.2% of NaCl to 1.33 M equal to 8% of NaCl) represents the natural conditions faced by *P. bermudensis* [13]. The average salinity in open oceans is ca. 0.6 M. Higher salinities can be found in regions with high evaporation (e.g., Mediterranean Sea and Red Sea). In contrast, hyposaline conditions occur in regions with high freshwater influx (e.g., Black Sea and Baltic Sea) [17]. We also investigated the zwitterionic metabolites in *P. bermudensis* in the total absence of NaCl in the medium. Surprisingly, the bacterium remained alive in the NaCl-free medium and still produced osmolytes, suggesting *P. bermudensis* possesses adaptative traits to cope with extreme environments.

Among the sulfur-containing zwitterionic metabolites, DMSP reached the highest intracellular concentrations (data are normalized to cell counts obtained with flow cytometry in Appendix A). DMSP was found in concentrations between 0.013 ± 0.0058 fmol cell^−1^ in the absence of NaCl and 0.3 to 0.4 fmol cell^−1^ at NaCl concentration of 0.66 M and above (Figure 1). This increase of the internal organic salt DMSP with increasing salinity suggests a role in osmoadaptation. Similar trends were also reported in *Vibrio parahaemolyticus* and several other *Vibrio* species [14]. DMSP might play a role in osmoadaptation under extreme salinity conditions as it was also observed in phytoplankton [17]. DMSP is thus not only a well-known osmoprotectant in phytoplankton, but also upregulated in bacteria under high salinity [6,19]. The other sulfur-containing zwitterions gonyol, cysteinolic acid, and DMSOP were substantially lower concentrated, and only gonyol was also found at increased concentrations under high salinity (Figure 1). Interestingly, the concentrations of DMSP and DMSOP do not correlate. While DMSP reached the highest concentrations at a salinity of 0.66 M and above, DMSOP reached maximum concentrations at 0.1 M NaCl. Since DMSOP is the oxidative transformation of DMSP, this might indicate that it responds to oxidative rather than salinity stress [11]. This result is supported by the maximum concentration of ROS in *P. bermudensis* under a NaCl concentration of 0.1 M (Figure 2).

The nitrogen-containing zwitterionic metabolites GBT and ectoine exceeded the amount of DMSP up to an order of magnitude. The metabolites that reached concentrations of up to 6 fmol cell^−1^ were strongly upregulated under high salinity. GBT and its precursor choline [20] showed a similar trend at all salinities (Figure 3). In most microbes, GBT is derived from the oxidation of choline [21]. GBT is a well-known osmoprotectant detected in several halophilic bacteria, such as *Actinopolyspora halophila* and *Halorhodopira halochloris* as well as in *E. coli* [7]. In contrast, GBT was not correlated with the salt concentration in the halophilic bacterium *Tetragenococcus halophila* where the compound was dominant at both high and low salinities [22]. Ectoine is described as an osmolyte produced by the halophilic bacterium *Halomonas elongata* and also accumulates in the diatoms *Thalassiosira weissflogii* and *Phaeodactylum tricornutum* under high salinity [7,23,24]. Its function as an osmolyte has been proven by the addition of elevated concentrations to cultures of *Streptomyces coelicolor* where it protected the bacterium from extreme salt stress [25]. This is in accordance with our studies, where high salt stress induces ectoine production.

The salinity dependence of the less abundant metabolites was complex. Some, such as gonyol and choline, also increased in cellular concentration with higher salinity. Gonyol, which is for the first time described in marine bacteria, reached the highest concentration at high salinity. It thus potentially plays a role in osmoadaptation as well. Gonyol is an inhibitor of DMSP lyases and is considered a means for algae to interfere with DMSP utilization by bacteria [17]. Whether the compound plays a similar role in bacteria remains to be elucidated.

Cysteinolic acid is not detectable at elevated salinity or in the absence of NaCl. It reaches its maximum concentration at the lowest tested salinity of 0.03 M (Figure 1). This indicates a hitherto unrecognized physiological role of the compound but no connection to osmoregulation. In contrast, the concentration of cysteinolic acid in the cells of phytoplankton *T. weissflogii* increased significantly when grown in a medium with elevated salinity [3].

Sarcosine (monomethylglycine), an intermediate in GBT biosynthesis [21], displayed a different pattern. Sarcosine showed the highest concentration around 0.30 ± 0.035 fmol cell^−1^ at a NaCl concentration of 0.23 M. Sarcosine is one of the common osmolytes in some halophilic bacteria due to its response to high salinity [6]. However, our data suggests that sarcosine is not, or only to a minor extent, involved in the osmotic adaptation of *P. bermudensis*.

Carnitine accumulated at both high and low salinity. Similarly, carnitine also accumulated in the bacterium *Tetragenococcus halophila* at both low and high salinities [21]. Carnitine is widely found in all living organisms and plays various roles, including protection against salt stress [22,23]. Proline accumulated at low salinity stress, particularly at a NaCl concentration of 0.23 M and at the highest salinity stress (NaCl concentration of 1.33 M). Unlike proline, the concentration of 4-hydroxyproline remained constant at a NaCl concentration of 0.33–0.66 M then gradually decreased by increasing and decreasing salinities.

In conclusion, the significant increase in GBT, choline, DMSP, ectoine, and gonyol at elevated salinity accounts quantitatively for most of the observed changes in the osmolyte composition of *P. bermudensis*. Whether the other detected metabolites contribute to adjustments at intermediate salinity will have to be proven in future experiments. It can only be speculated why the bacterium relies on the regulation of an entire group of highly polar metabolites and not one master regulator. Since fundamentally different metabolic pathways fuel the pool of polar metabolites, the high number of upregulated metabolites might alleviate metabolic bottlenecks occurring if only one or a few metabolites were involved in regulation. Further, the involvement of different metabolic pathways might allow for a better fine-tuning of metabolic responses. However, this point’s final clarification might only be obtained by future knock-out experiments.

### 2.2. Response to Temperature Variation

The examined temperature regime of 18 °C, 28 °C, and 38 °C represents a selection of conditions encountered by the mesophilic halophilic bacterium *P. bermudensis* in natural environments [13]. *P. bermudensis* is widely distributed in oceans and found under diverse environmental conditions and thus needs efficient adaptation strategies. The bacterium grows and survives in a temperature range between 10 °C and 40 °C with an optimum of 30–33 °C [13]. In accordance, growth rates in our experiments were high at 28 °C and 38 °C and lower at 18 °C. The intracellular amount of DMSOP and DMSP increased around 28- and 35-fold, respectively, from 18 °C to 38 °C (Figure 4). In contrast to the dysregulated behavior of the two biosynthetically derived metabolites under salinity changes, the temperature change affected both metabolites similarly. This suggests that DMSOP production from DMSP might function as an antioxidant mechanism under heat stress. This is also supported by the 8-fold higher ROS concentration at 38 °C compared to 18 °C (Figure 5). Heat stress can induce oxidative stress in some bacteria and phytoplankton [26]. Thus, for example, for the phytoplankton *Alexandrium minutum*, temperature-dependent DMSP increase was interpreted as an antioxidant mechanism in response to oxidative stress in phytoplankton [27]. Furthermore, intracellular DMSP and DMSOP also increased in response to oxidative stress in the late-stationary-phase cultures of the alga *Isochrysis galbana* [11].

Intracellular gonyol and cysteinolic acid were lowest at 28 °C. The cellular concentration of gonyol increased ca. 4-fold and cysteinolic acid around 10-fold when the temperature was increased from 28 °C to 38 °C. Upon reduction of the temperature to 18 °C, gonyol and cysteinolic acid more than doubled. Cysteinolic acid is also upregulated in the green macroalga *Ulva mutabilis* during cold stress from 18 to 2 °C [28]. However, the overall concentrations of these metabolites are substantially lower than those of DMSP, suggesting rather specific roles in the physiology of *P. bermudensis*.

The response of nitrogen-containing zwitterionic compounds to temperature changes can be classified into three groups. The biosynthetically related GBT, choline, and sarcosine levels were lowest at the intermediate temperature of 28 °C. The highest concentrations were found at 38 °C, while levels in between the extremes are observed at 18 °C (Figure 6). GBT and choline also increase at higher temperatures in *E. coli* where they protect β-galactosidase and citrate synthase against thermodenaturation while triggering citrate synthase renaturation [29]. GBT and choline were also identified as thermoprotectants during cold stress in Bacillus subtilis [30].

4-hydroxyproline and carnitine concentrations increased with a temperature increase from 18 °C to 38 °C. Similar to the DMSP response to ROS production at elevated temperatures, we suggested that carnitine and 4-hydroxyproline acted as antioxidants during thermal stress in *P. bermudensis*. Indeed, carnitine and 4-hydroxyproline are widely known as potent antioxidants in human and animal cells [31,32]. A previous study also reported that carnitine acts as a potent antioxidant by preventing lipid oxidation after thermal stress exposure in rats [33].

Ectoine and proline followed similar trends. These compounds are abundant at low temperatures (18 °C). Ectoine is widely known as both a cold and heat stress protectant in bacteria and some *Archaea* [34]. In our study, ectoine most likely plays a substantial role only at the colder temperature of 18 °C. In contrast to our study, the bacterium *Streptomyces coelicolor* harbors high ectoine concentrations under heat stress at 39 °C [25].

### 2.3. Metabolic Response of P. bermudensis after Exposure to an Algal Extract

To determine the effect of metabolites from associated algae on the production of zwitterionic metabolites in *P. bermudensis*, the bacteria were incubated with an algal extract. Therefore, an axenic T. striata culture was extracted, and the extract was added to the bacterial cultures with a final concentration of 0.15 μg μL^−1^, 0.30 μg μL^−1^, 0.60 μg μL^−1^, or 1.2 μg μL^−1^ of dried extract. Concentrations of zwitterionic metabolites in the extract are given in Appendix A. *T. striata* occurs in association with *P. bermudensis*, and the bacterium induces growth and increase in the biomass of the alga [15,16]. Despite this close connection, no significant changes in zwitterionic metabolites were observed between the control and treatments (Appendix A). T. striata also contains zwitterionic metabolites, including all compounds found in *P. bermudensis* in this study except gonyol. Even if detailed conclusions about the uptake of minor amounts of metabolites could only be drawn if isotopically labeled substrates were added, our finding suggests that the bacterium does not take up and store substantial amounts of metabolites from the algae. Further, it cannot be excluded that the bacteria rapidly consume and transform taken-up substrates. Such fine-tuned interactions would need a separate detailed study using isotopically labeled substrates and more sampling time points to account for potential physiological changes in the different growth phases of the bacteria.

## 3. Materials and Methods

### 3.1. Identity of Pelagibaca bermudensis

*P. bermudensis* HTCC2601 was purchased from the National Center for Marine Algae and Microbiota at Bigelow Laboratory, USA (https://ncma.bigelow.org/ncma-b17) (Accessed on 24 June 2021). Before the experiment, bacterial identity was confirmed by the following method [13,35]: The bacterium was cultivated on marine broth agar plates. After incubation for 3 days at 28 °C, a single colony was transferred into Eppendorf tubes with 100 µL TE buffer, then heated at 100 °C for 5 min, put on ice for 5 min, followed by centrifugation at 12,000 rpm for 5 min at room temperature. The collected cell pellet was used for DNA extraction. A 0.5 µL properly diluted DNA sample was added as the template to 20 μL PCR reaction solution with 1.25 U DreamTaq DNA polymerase (Thermo Scientific, Dreieich, Germany). The 16S ribosomal RNA (16S rRNA) gene was amplified with the universal primers 27F (5′-AGAGTTTGATCCTGGCTCAG-3′) and 1492R (5′-ACGGHTACCTTGTTACGACTT-3′) under the PCR conditions: (i) 5 min at 95 °C, (ii) 30 cycles of 30 s at 95 °C, (iii) 30 s at 55 °C, (iv) 90 s at 72 °C, (v) 10 min at 72 °C for extension then paused at 24  °C. The PCR products were verified by agarose gel electrophoresis for quality control. The PCR products were submitted to Eurofins Genomics (Eurofins Genomics, Ebersberg, Germany) for sequencing. The sequence was submitted to BLAST (https://blast.ncbi.nlm.nih.gov/Blast.cgi) (Accessed on 2 August 2021). The result showed that the sequence was fully matched with *Salipiger bermudensis* HTCC2601 (now classified as *Pelagibaca bermudensis*) [13,36].

### 3.2. Cultivation of Pelagibaca bermudensis

*P. bermudensis* HTCC2601 was cultivated on marine broth agar plates. After incubation for 3 days at 28 °C, a single colony was transferred into a marine broth medium (Carl Roth, Karlsruhe, Germany) in tissue-culture flasks according to a previously published protocol [11]. The cultures were grown under shaking (130 rpm). Bacterial cultures grown with 0.33 M NaCl at 28 °C served as control. All media were autoclaved before use.

### 3.3. Cultivation of Tetraselmis striata

Axenic culture of *T. striata* RCC131 was purchased from the Roscoff Culture Collection, France (https://roscoff-culture-collection.org/rcc-strain-details/131) (Accessed on 29 October 2021). *T. striata* RCC131 was cultivated in K medium without silicate according to Keller and Guillard [37,38]. Cultivation was started from a stationary phase stock culture which was diluted 20-fold with fresh medium and cultivated again to the late exponential phase (determined in independent preliminary experiments recording cell counts and content of chlorophyll a) before the extraction. All cultures grown in 100 mL polystyrene cell culture bottles with membrane filter screw caps for gas exchange were maintained based on a modified method from Patidar et al. at a temperature of 18 °C ± 2 °C with 14:10 light–dark cycle with light provided by Osram biolux lamps (40 µmol photons m^−2^s^−1^ between 400 and 700 nm) [12]. All cultures were cultivated in three biological replicates.

Axenic culture of *T. striata* was obtained according to Shishlyannikov et al. [39]. Xenic culture of *T. striata* was filtered through pluriStrainer^®^ (cell strainer) with a pore size of 5 µm. Algal cells obtained by the strainer were washed with sterile K medium. Triton X-100 (Sigma-Aldrich, St. Louis, MO, USA) was added to the culture in the tissue flask with K medium to a final concentration of 20 µg mL^−1^. It was shaken for 30 s and immediately filtered through a pluriStrainer^®^ (cell strainer). The cells obtained by the strainer were washed with sterile K medium and transferred to K medium in a tissue flask. The culture was treated with ciprofloxacin 25 µg mL^−1^ and erythromycin 12.5 µg mL^−1^ and incubated for 24 h. Then cells were washed and transferred into sterile K medium. Axenicity was periodically checked by microscopy, flow cytometry, and by plating aliquots of each culture on marine broth agar plates. The selected antibiotics were determined based on prior antibiotic susceptibility tests by measuring the MIC toward some bacteria in 96-well plates according to Wiegand et al. [40].

*T. striata* was extracted at the late exponential phase of growth by filtration of 40 mL in three replicates under reduced pressure (Whatman GF/C grade microfiber filters) of 500 mbar. The filter was immediately transferred into 4 mL glass vials containing 500 µL of methanol, while another 500 µL of methanol was added directly to the filter and vortexed for 30 s. Samples were dried under a nitrogen stream. The dried extracts were used for further incubation of *P. bermudensis.*

### 3.4. Stress Treatments

#### 3.4.1. Salinity Treatment

To study the effect of salinity stress, the salinity of the medium was adjusted by modifying the content of NaCl, resulting in NaCl concentrations of 0.033 M, 0.10 M, 0.17 M, 0.23 M, 0.33 M, 0.50 M, 0.66 M, 1.0 M, 1.33 M, and absence of NaCl. The concentration of other nutrients remained the same. All media were autoclaved before use. The control condition was 0.33 M at 28 °C. All media were autoclaved before use.

Standing cultures in marine broth medium (Carl Roth) were kept in tissue-culture flasks according to a previously published study [11]. Starting from a stock culture of *P. bermudensis*, treatments were adapted to the different salinities by cultivating in each respective media starting with an OD_600_ of 0.2. The bacterial cultures were then grown to their late exponential phase. Then, these cultures were used for inoculation of three biological replicates by diluting with their respective media to an OD_600_ of 0.2. The bacterial cultures were grown until their late exponential phase. Bacterial growth was recorded by measuring absorbance (OD_600_) using a Spectro-photometer. The cultures were grown under shaking (130 rpm). The sampling for osmolyte quantification was in the late exponential growth phase. All cultures were cultivated in three biological replicates. Each medium was used as a blank control for analytical analysis.

#### 3.4.2. Thermal Stress Treatment

The cultures were grown under shaking (130 rpm) at 18 ± 2 °C, 28 ± 2 °C, and 38 ± 2 °C. All bacterial cultures were cultivated at 0.33 M NaCl concentration. Bacterial growth was recorded by measuring OD_600_ using a Spectro-photometer. The sampling points for osmolyte quantification were selected during the late exponential growth phase of cultures. All cultures were cultivated in three biological replicates.

#### 3.4.3. Incubation with Extract of *T. striata*

*P. bermudensis* HTCC2601 was cultivated as standing culture in normal marine broth medium (Carl Roth) with starting OD_600_ of 0.2. The bacterial cultures were supplemented with *T. striata* extract (dissolved in marine broth medium with 0.33 M NaCl) at final concentrations of 0.15 µg μL^−1^, 0.30 µg μL^−1^, 0.60 µg μL^−1^, and 1.2 µg μL^−1^ extract. The cultures were grown under shaking (130 rpm) at 28 ± 2 °C to the late exponential phase (OD_600_ of 1.90). Bacterial cultures without the addition of algal extract were used as a control. Bacterial growth was recorded by measuring absorbance (OD_600_) using the spectrophotometer. Sampling points for osmolyte quantification were selected from the late exponentially growing cultures. All cultures were cultivated in three biological replicates.

### 3.5. Cell Counting

To determine the final cell densities, 25 µL aliquots of cultures were analyzed on a BD AccuriTM C6 flow cytometer. The discriminator was set to forward light scatter. Before data collection, the instrument was validated using diluted beads solutions with a known concentration. Samples were stained with SYBR Gold dye (SYBR. Gold Nucleic Acid Gel Stain (10,000× Concentrate in DMSO), Thermo Scientific, 10,000-fold diluted from stock solution) for 15 min in the dark according to previous standard protocols [41,42]. Then the samples were measured with the following settings: 25 µL per sample, 4 µL min^−1^, and 3 washing steps between each sample with milli-Q water.

### 3.6. ROS Staining

The detection of intracellular ROS was performed according to Grave et al. and Roesslein et al. [43,44] using a fluorometric microplate assay by detecting oxidation of 2′,7′-dichlorofluorescin-diacetate (DCF-DA) to the highly fluorescent compound 2′,7′ dichlorofluorescein (DCF) in the presence of reactive oxygen species (ROS). A stock solution of DCF-DA was reconstituted in DMSO (Sigma-Aldrich, St. Louis, MO, USA) to a concentration of 1 mM and stored at 4 °C. Aliquots of the bacterial cultures (100 μL) at their late exponential phase were collected. Cells were then washed twice with ice-cold phosphate-buffered saline (PBS). DCF-DA was added to the samples at a final concentration of 25 µM. Samples were incubated for 60 min at 37 °C under dark conditions. Cells were then washed with phosphate-buffered saline (PBS) to remove the excess dye. The cells were transferred to a 96-well plate. Intracellular DCF fluorescence was measured on a plate reader using an excitation maximum of 488 nm and an emission maximum of 535 nm. The fluorescence intensity is equal to the intracellular ROS levels. Samples were run in three biological replicates. The fluorescence intensity obtained from the samples without dye was subtracted as blanks. The DCF fluorescence of each sample was normalized by normalization based on the ratio of their cell count. PBS added with DCF-DA was used as a negative control.

### 3.7. Sample Preparation for MS-Analysis

Aliquots of the bacterial cultures (100 μL) were centrifuged for 25 min at 16,100 g and the supernatant was removed by pipetting. The pellets were taken up in 100 μL of a mixture of acetonitrile and water (9:1 *v*/*v*) containing an aqueous solution of an internal standard mixture (D_6_-DMSA, D_6_-DMSP, D_3_-ectoine, and D_3_-gonyol with final concentration of 500 nM) and vortexed for 30 s. Cells were disrupted by sonication using 6 cycles, 10 s pulses with 40% intensity in a Bandelin Sonoplus ultrasound homogenizer (Bandelin). The samples were again centrifuged for 10 min at 16,100 g and 5 µL of the supernatants were directly submitted to UHPLC/HRMS for analysis.

### 3.8. Equipment

Analytical separation and quantification were completed using a Dionex Ultimate 3000 system (Thermo Scientific, Dreieich, Germany) connected to a Q-Exactive Plus Orbitrap mass spectrometer (Thermo Scientific, Dreieich, Germany). The UHPLC column was a Sequent ZIC-HILIC column (2.1 × 150 mm, 5 µm) coupled with SeQuent ZIC HILIC guard column (2.1 × 20 mm, 5 µm) (Merck, Darmstadt, Germany). MS data were processed using the Xcalibur software. Electrospray ionization was performed in positive mode ionization with the following parameters: capillary temperature, 380 °C; spray voltage, 3000 V; sheath gas flow, 60 arbitrary units; and aux gas flow, 20 arbitrary units.

### 3.9. Osmolyte Analysis

For the separation of osmolytes via UHPLC, we used the method of Thume et al. [11]. The mobile phase consisted of high-purity water (Th Geyer GmbH, Renningen, Germany) with 2% acetonitrile LC-MS grade (Th Geyer GmbH, Renningen, Germany) and 0.1% formic acid LC-MS grade (Thermo Scientific, Dreieich, Germany) (solvent A) and 90% acetonitrile with 10% 1 mmol of aqueous ammonium acetate LC-MS grade (LGC Promochem, Wesel, Germany) (solvent B). A gradient elution was performed using isocratic elution of 100% solvent B for 1 min, followed by a linear gradient from 100% solvent B to 20% solvent B for 5.5 min and a linear gradient from 20% solvent B to 100% solvent B for 0.6 min, and isocratic equilibration at 100% solvent B for 2.9 min. The total run time was 10 min. The column was kept at 25 °C. The flow rate was set at 0.6 mL min^−1^. The injection volume was 5 µL. Full scan mode was set from 75 to 200 *m/z* at a resolution of 70,000. Before running the samples, the UHPLC was controlled by repeatedly running blanks.

Identification of osmolytes was carried out by comparing MS and MS/MS of osmolytes in the samples with commercial and synthetic standards. Commercially available standards used were l-carnitine (Sigma-Aldrich, St. Louis, MO, USA), Choline (Sigma-Aldrich, St. Louis, MO, USA), sarcosine (ABCR GmbH, Karlsruhe, Germany), l-ectoine (Sigma-Aldrich, St. Louis, MO, USA), l-proline (Sigma-Aldrich, St. Louis, MO, USA), l-4-hydroxyproline (Fluka analytical, Steinheim, Germany). Other standards were obtained by synthesis as described in previous studies [11,17,23]. MS/MS fragmentation was performed with a collision energy of 35 V. The spectral charts were made using the Metabolomics Spectrum Resolver (https://metabolomics-usi.ucsd.edu/) (Accessed on 13 August 2022) according to the method from Bittremieux et al. [45] (Appendix A).

For quantification, a calibration curve of the standards was recorded followed by comparisons of the peak area of the analytes with the calibration curve of the internal standard [17]. As internal standards for quantification of DMSP and DMSOP, we used D_6_-DMSP and D_6_-DMSA, respectively. As internal standard for the quantification of ectoine, we used D_3_-ectoine. For all other osmolytes, D_3_-gonyol was used as the internal standard. D_6_-DMSP, D_6_-DMSA, D_3_-gonyol, and D_3_-ectoine were obtained by synthesis in our laboratory based on a published procedure [6,18].

For DMSOP, the calibration curve (*n* = 3) for the area of the molecular ion was y = 2.16 × 10^−4^ *x* with r = 0.9975, limit of detection (LOD) = 22.2 nM, limit of quantification (LOQ) = 78.31 nM; for DMSP, the calibration curve (*n* = 3) for the area of the molecular ion was y = 1.27 × 10^−3^ *x* with r = 0.9973, limit of detection (LOD) = 27.6 nM, limit of quantification (LOQ) = 96.5 nM; for gonyol y = 7.74 × 10^−4^ *x* with r = 0.9583, limit of detection (LOD) = 1.19 nM, limit of detection (LOD) = 4.23 nM; for cysteinolic acid y = 1.10 × 10^−3^ *x* with r = 0.9938, limit of detection (LOD) = 0.79 nM, limit of quantification (LOQ) = 2.83 nM; for L-ectoine, y = 6.21 × 10^−3^ *x* with r = 0.9985, limit of detection (LOD) = 15.7 nM, limit of quantification (LOQ) = 56.0 nM; for l-carnitine, y = 6.13 × 10^−3^ *x* with r = 0.9978, limit of detection (LOD) = 22.3 nM, limit of quantification (LOQ) = 78.66 nM; for choline, y = 4.99 × 10^−3^ *x* with r = 0.9951, limit of detection (LOD) = 56.0 nM, limit of quantification (LOQ) = 197 nM; for glycine betaine, y = 1.39 × 10^−2^ *x* with r = 0.9919, limit of detection (LOD) = 57.2 nM, limit of quantification (LOQ) = 201 nM; for sarcosine y = 4.27 × 10^−3^ *x* with r = 0.9951, limit of detection (LOD) = 47.9 nM, limit of quantification (LOQ) = 169 nM; for l-proline y = 1.29 × 10^−2^ *x* with r = 0.9937, limit of detection (LOD) = 46 nM, limit of quantification (LOQ) = 162 nM; for l-4-hydroxyproline y = 4.72 × 10^−3^ *x* with r = 0.9927, limit of detection (LOD) = 48.3 nM, limit of quantification (LOQ) = 169 nM. All calibration curves passed Mendel’s test.

## 4. Conclusions

We describe here the monitoring of eleven highly polar compounds (DMSOP, DMSP, gonyol, cysteinolic acid, ectoine, glycine betaine (GBT), carnitine, sarcosine, choline, proline, and 4-hydroxyproline) during a stress response in the halophilic heterotrophic bacterium *P. bermudensis*. We clearly show that orchestrated responses to osmotic stress and temperature variation can be observed among this compound class. No response to algal extracts indicates that the metabolites are physiologically controlled by the bacterium and not taken up in an uncontrolled manner from the medium.

## Figures and Tables

**Figure 1 marinedrugs-20-00727-f001:**
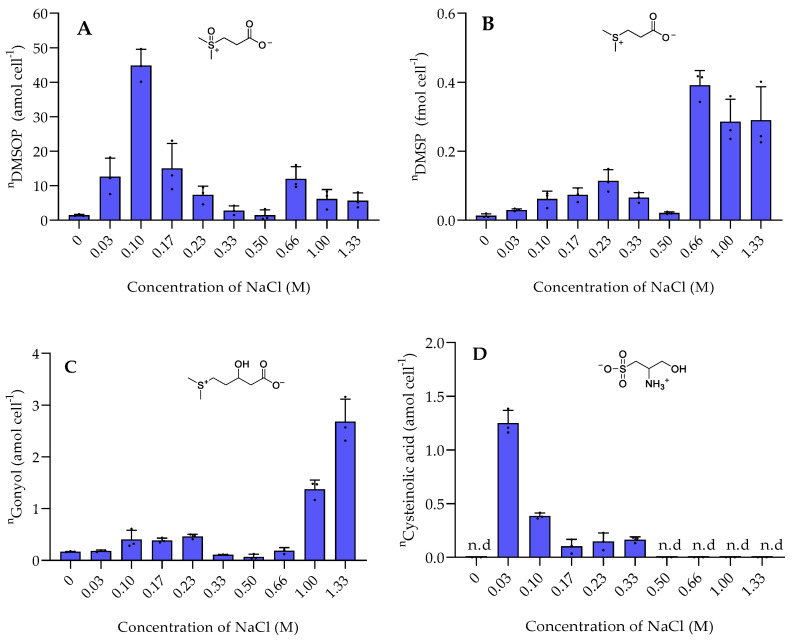
The intracellular concentration of sulfur-containing zwitterionic metabolites of *P. bermudensis* HTCC2601 under salinity stress at late exponential growth. (**A**) DMSOP, (**B**) DMSP, (**C**) gonyol, and (**D**) cysteinolic acid. Concentrations are normalized to cell counts. Error bars represent standard deviation (biological replicates, *N* = 3). n.d. = not detected—the compound concentration was below the limit of detection (LOD). ^n^ = amount per cell.

**Figure 2 marinedrugs-20-00727-f002:**
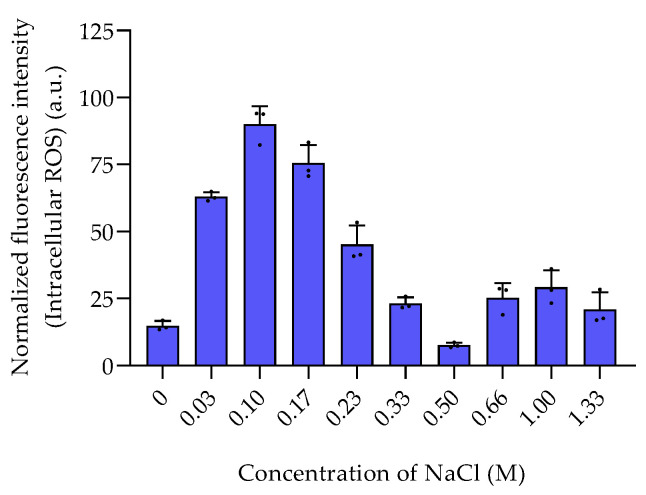
The intracellular ROS levels of *P. bermudensis* HTCC2601 under salinity stress. The fluorescence intensity of each sample was normalized to the cell count. Error bars represent standard deviation (biological replicates, *N* = 3). (a.u.): arbitrary unit.

**Figure 3 marinedrugs-20-00727-f003:**
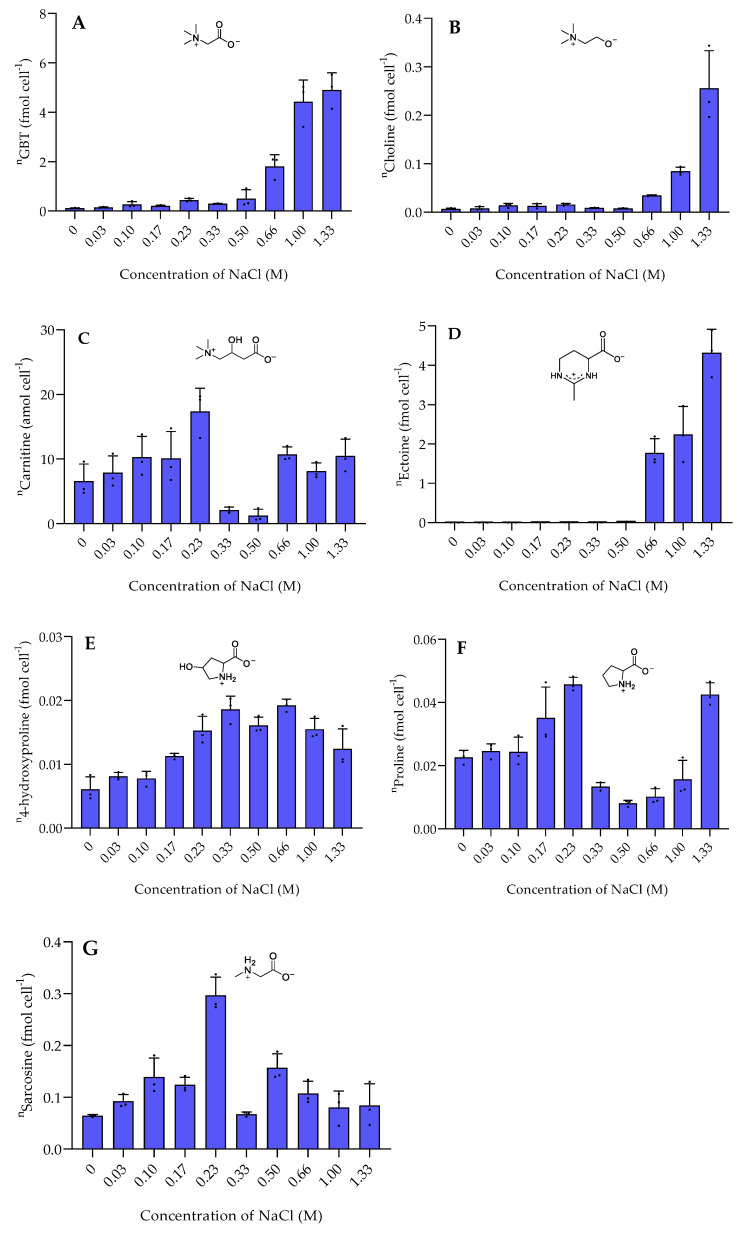
The intracellular concentration of nitrogen-containing zwitterionic metabolites of *P. bermudensis* HTCC2601 under salinity stress at late exponential growth. (**A**) Glycine betaine, (**B**) Choline, (**C**) carnitine, (**D**) ectoine, (**E**) 4-hydroxyproline, (**F**) proline, and (**G**) Sarcosine. Concentrations are normalized to cell count and error bars represent standard deviation (biological replicates, *N* = 3). ^n^ = amount per cell.

**Figure 4 marinedrugs-20-00727-f004:**
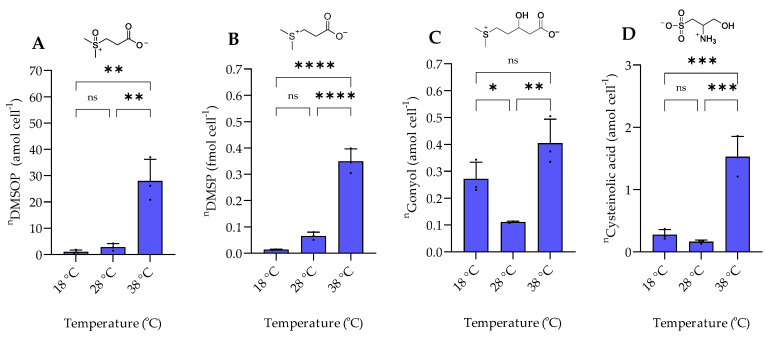
The intracellular concentration of sulfur-containing zwitterionic metabolites of *P. bermudensis* HTCC2601 under temperature stress at late exponential growth. (**A**) DMSOP, (**B**) DMSP, (**C**) gonyol, and (**D**) cysteinolic acid. Concentrations are normalized to cell count. Error bars represent standard deviation (biological replicates, *N* = 3). Statistical analysis is based on one-way ANOVA with a Tukey test for multiple comparison procedures (*p*-value: * *p* < 0.05, ** *p* < 0.01, *** *p* < 0.001, **** *p* < 0.0001, ns: not significant). ^n^ = amount per cell.

**Figure 5 marinedrugs-20-00727-f005:**
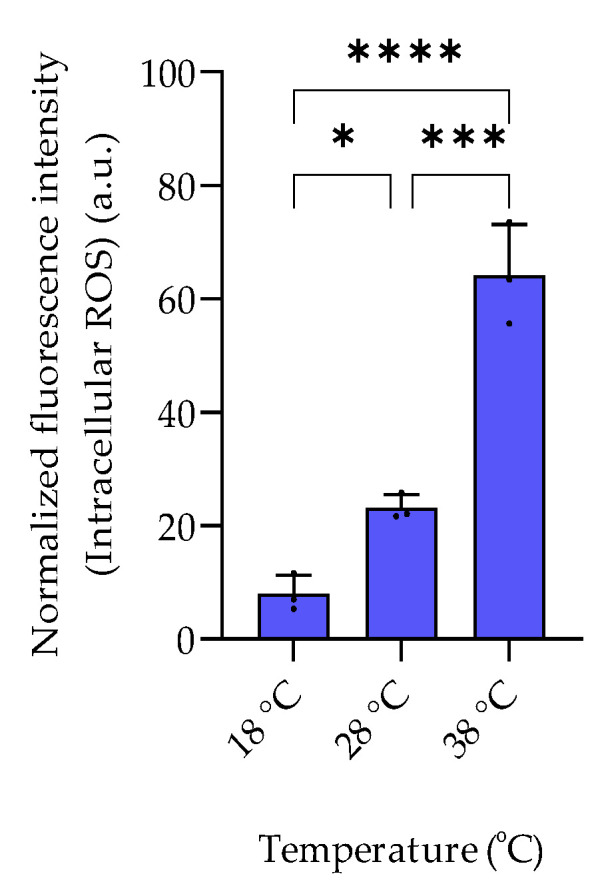
Temperature-dependent intracellular ROS levels of *P. bermudensis* HTCC2601. The fluorescence intensity of each sample was normalized to the cell count at late exponential growth. Error bars represent the standard deviation (biological replicates, *N* = 3). Statistical analysis is based on one-way ANOVA with a Tukey test for multiple comparison procedures. * *p* < 0.05, *** *p* < 0.001, **** *p* < 0.0001. a.u.: arbitrary unit.

**Figure 6 marinedrugs-20-00727-f006:**
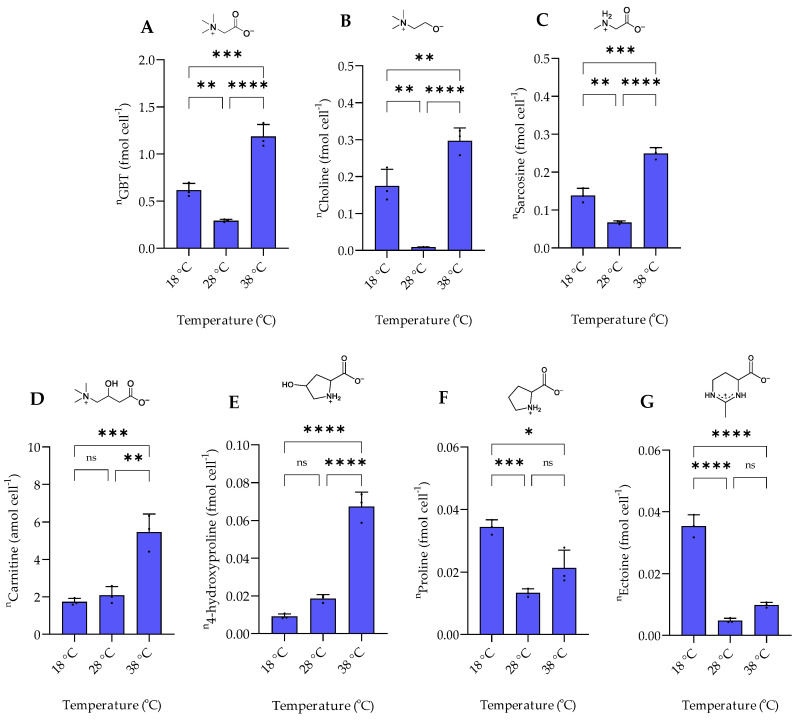
The intracellular concentration of nitrogen-containing zwitterionic metabolites of *P. bermudensis* HTCC2601 under temperature stress at late exponential growth. (**A**) Glycine betaine, (**B**) choline, and (**C**) sarcosine showed a similar trend, while (**D**) carnitine showed a similar trend with (**E**) 4-hydroxyproline. (**F**) Proline showed a similar trend with (**G**) ectoine. Concentrations are normalized to cell count. Error bars represent standard deviation (biological replicates, *N* = 3). Statistical analysis is based on one-way ANOVA with a Tukey test for multiple comparison procedures. * *p* < 0.05, ** *p* < 0.01, *** *p* < 0.001, **** *p* < 0.0001, ns: not significant. ^n^ = amount per cell.

## Data Availability

Not applicable.

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
