# Peer review of "Orchestrated Response of Intracellular Zwitterionic Metabolites in Stress Adaptation of the Halophilic Heterotrophic Bacterium Pelagibaca bermudensis"

_marinedrugs, 2022, doi:10.3390/md20110727_

Round 1

Reviewer 1 Report

The authors investigated the effects of salinity, temperature and Tetraselmis extract on the biosynthesis of 11 different osmolytes by a bacterium. Use of UHPLC with HREIMS for the detection of these osmolytes is impressive. Results provided some idea in terms of how these metabolites responded to various stresses. 

The text mentions the bacterium as Pelagibacter bermudensis. Did the authors mean Pelagibaca bermudensis (https://ncma.bigelow.org/ncma-b17 )? Please check the correct genus name for the strain. 

Various osmolytes seem to be synthesized together for a given stress factor (e.g. high salinity). I would have liked to see more discussion on why this bacterium needed various osmolytes for a specific stress factor and what their functions might be. 

The authors also suggest that some osmolytes respond to oxidative stress. It would have been more convincing if the authors exposed the bacteria to some oxidative stress chemicals to see if they also induced osmolyte biosynthesis similar to the stress factor they applied. 

In summary I believe the text might improve if the authors speculate on the possible functions of various osmolytes.

Author Response

The authors investigated the effects of salinity, temperature and Tetraselmis extract on the biosynthesis of 11 different osmolytes by a bacterium. Use of UHPLC with HREIMS for the detection of these osmolytes is impressive. Results provided some idea in terms of how these metabolites responded to various stresses. 

We thank the referee for these highly exciting positive comments.

The text mentions the bacterium as Pelagibacter bermudensis. Did the authors mean Pelagibaca bermudensis (https://ncma.bigelow.org/ncma-b17)? Please check the correct genus name for the strain. 

Thank you. We corrected every mention of Pelagibacter bermudensis into Pelagibaca bermudensis in the manuscript - we apologize for this embarrassing mistake.

Various osmolytes seem to be synthesized together for a given stress factor (e.g. high salinity). I would have liked to see more discussion on why this bacterium needed various osmolytes for a specific stress factor and what their functions might be. The authors also suggest that some osmolytes respond to oxidative stress. It would have been more convincing if the authors exposed the bacteria to some oxidative stress chemicals to see if they also induced osmolyte biosynthesis similar to the stress factor they applied. In summary I believe the text might improve if the authors speculate on the possible functions of various osmolytes.

Thank you for your suggestion. In this manuscript we mainly focus on the effect of salinity stress, thermal stress, and exposure to an algal extract toward osmolytes production. The idea why we measured ROS concentration under salinity stress on this study is because we found that DMSOP reaches maximum concentrations at 0.1 M NaCl. Since DMSOP is the oxidative transformation of DMSP, we wanted to test the hypothesis that it might respond to oxidative rather than salinity stress. This result is supported by the maximum concentration of ROS in P. bermudensis under a NaCl concentration of 0.1 M. We felt that this result stands for itself and does not require additional manipulation with oxidants.

In addition, we also measured ROS concentration during the thermal stress because we would like to support our hypothesis that DMSOP production from DMSP might function as antioxidant mechanism at elevated temperature since the heat stress can induce oxidative stress in some bacteria. This is supported by the 8-fold higher ROS concentration at 38 °C compared to 18 ˚C.

We now expand the discussion as suggested by the referee (especially line 199ff).

Reviewer 2 Report

This is a really well presented study demonstrating the intracellular regulation of a wide range of S- and N osmolytes under salinity and temperature stress. It is common to see one or two compounds addressed in a study like this. These authors have analysed 11 zwitterionic compounds of interest, which makes this a comprehensive piece of work. The quality of the study is high. I have outlined some mainly minor comments below.

My most major comment is around the conclusion of P. bermudensis not taking up and storing metabolites from an experiment where the metabolites were not quantified in the algal extract (or the data were not shown to demonstrate this). This data needs to be added to the supplementary information. Further, this aspect of the study may well be floored, as literature demonstrates that algal derived DON is rapidly consumed by bacteria (Polimene et al. 2017 PLoS ONE 12(2), e0171391). The sampling point of the bacterial cultures isn’t indicated beyond “late exponential”. The possible high rate of uptake of organic material from the T. striata extracts in the early hours and days of growth needs to be considered. Further, without the addition of labelled substrates, it cannot be shown that the bacterium isn’t using a combination of transport and biosynthesis to maintain an optimum intracellular concentration. The results and discussion should be edited to reflect these points.

Minor comments:

Line 61: Reads as if a word is missing after “major”, or remove major altogether.

Line 122: spelling observed.

Line 123-124: This needs clarifying – how does observation of DMSP in phytoplankton suggest a role for osmoadaptation under extreme salinity conditions?

Figures 1, 3, 4 and 6 have superscript n in the y axis label. This is not defined – does it represent normalized?

Line 179: Needs rephrasing; “did not contribute to most alteration in the osmotic condition” doesn’t make sense.

Line 213: Insert comma after minutum

Line 240: Spelling of sarcosine

Line 248: Suggest change to “acted as antioxidants”

Line 255-256: Suggest change to: “ectoine most likely plays a substantial role only at the colder temperature of 18 oC.”

Line 272-273: Suggest move “of the dried extract” to the end of the sentence for it to make sense.

Line 352: Suggest replace “till” with “until”

Line 362: exponential not exponentially

Line 373: Use were in place of was

Author Response

This is a really well presented study demonstrating the intracellular regulation of a wide range of S- and N osmolytes under salinity and temperature stress. It is common to see one or two compounds addressed in a study like this. These authors have analysed 11 zwitterionic compounds of interest, which makes this a comprehensive piece of work. The quality of the study is high. I have outlined some mainly minor comments below.

We thank the referee for these highly encouraging positive comments

My most major comment is around the conclusion of P. bermudensis not taking up and storing metabolites from an experiment where the metabolites were not quantified in the algal extract (or the data were not shown to demonstrate this). This data needs to be added to the supplementary information. Further, this aspect of the study may well be floored, as literature demonstrates that algal derived DON is rapidly consumed by bacteria (Polimene et al. 2017 PLoS ONE 12(2), e0171391). The sampling point of the bacterial cultures isn’t indicated beyond “late exponential”. The possible high rate of uptake of organic material from the T. striata extracts in the early hours and days of growth needs to be considered. Further, without the addition of labelled substrates, it cannot be shown that the bacterium isn’t using a combination of transport and biosynthesis to maintain an optimum intracellular concentration. The results and discussion should be edited to reflect these points.

We fully agree and added the information about the concentration of zwitterionic metabolites in the algal extract added for incubation study in the Supplementary material (Table S1). We also expand the discussion on this chapter mentioning the limitations of the study raised by the referee (section 2.3).

Minor comments:

Line 61: Reads as if a word is missing after “major”, or remove major altogether.

Thank you. We corrected it

Line 122: spelling observed.

Thank you. Rephrased

Line 123-124: This needs clarifying – how does observation of DMSP in phytoplankton suggest a role for osmoadaptation under extreme salinity conditions?

Clarified: “This increase of the internal organic salt DMSP with increasing salinity suggests a role in osmoadaptation. Similar trends were also reported in Vibrio parahaemolyticus and several other Vibrio species [14].”

Figures 1, 3, 4 and 6 have superscript n in the y axis label. This is not defined – does it represent normalized?

“n represents amount per cell”. Now indicated in figure legends.

Line 179: Needs rephrasing; “did not contribute to most alteration in the osmotic condition” doesn’t make sense.

Re-phrased: “However, it suggests that sarcosine is not, or only to a minor extend, involved did not contribute to most alteration in the osmotic conditionadaptation, and played no role in osmoadaptation in P. bermudensis.”

Line 213: Insert comma after minutum

Thank you. We corrected it

Line 240: Spelling of sarcosine

Thank you. We corrected it

Line 248: Suggest change to “acted as antioxidants”

Thank you. We corrected it

Line 255-256: Suggest change to: “ectoine most likely plays a substantial role only at the colder temperature of 18 oC.”

Thank you. We corrected it

Line 272-273: Suggest move “of the dried extract” to the end of the sentence for it to make sense.

Thank you. We corrected it

Line 352: Suggest replace “till” with “until”

Thank you. We corrected it

Line 362: exponential not exponentially

Thank you. We corrected it

Line 373: Use were in place of was

Thank you. We corrected it

Reviewer 3 Report

The authors of the manuscript entitled “Orchestrated Regulation of Intracellular Zwitterionic Metabolites in Stress Adaptation of the Halophilic Heterotrophic Bacterium Pelagibacter bermudensis” aim to investigate the effect of several stress conditions including different salinities, temperatures, and the exposure to organic metabolites released by algae on the halophilic heterotrophic bacterium Pelagibacter bermudensis. Despite the complex analysis, The rationale of the work appears obscure why the author chose this bacterium. And the microalga Tetraselmis striata extracts? Many metheodological issues should be afforded therefore in its current form the manuscript is not acceptable for this prestigiousus journal. 

Some questions/suggestions are listed below:

Line 14 “organic metabolites released by algae” could the authors specify which algae they refer? 

Line 20 “for bacterial survival in challenging environments” which environments? Please be clear

Line 34 “cell organelles against various forms of stress” the sentence appears confused. Why do the authors cite organelles if the topic is the bacterium?

Line 60-61 the authors assessed that the Pelagibacter bermudensis “has genes encoding for the production of the major highly polar DMSP 61 and is also a DMSOP producer” why no genetic analysis was performed in this study? the entire manuscript appears out off-topic with respect to the title “ Orchestrated Regulation of Intracellular Zwitterionic Metabolites in Stress Adaptation of the Halophilic Heterotrophic Bacterium Pelagibacter bermudensis” No regulation analysis has been performed but only a response analysis to various conditions. 

Line 203 is not a temperature range the authors examine only three temperatures why? If the objective was to investigate the physiology of the P. bermidensis is strange exhamine a so far temperature “38°C” from the environmental and optimal temperature. Please could the authors clarify this point?

Line 272-274 the period is very confusing please rewrite

Author Response

The authors of the manuscript entitled “Orchestrated Regulation of Intracellular Zwitterionic Metabolites in Stress Adaptation of the Halophilic Heterotrophic Bacterium Pelagibacter bermudensis” aim to investigate the effect of several stress conditions including different salinities, temperatures, and the exposure to organic metabolites released by algae on the halophilic heterotrophic bacterium Pelagibacter bermudensis. Despite the complex analysis, The rationale of the work appears obscure why the author chose this bacterium. And the microalga Tetraselmis striata extracts? Many metheodological issues should be afforded therefore in its current form the manuscript is not acceptable for this prestigiousus journal. 

We motivate the investigation of P. bermudensis line 58 ff: “We selected the marine halophilic heterotrophic bacterium P. bermudensis as a study organism to elucidate the response of zwitterionic metabolites under stress. The marine halophilic bacterium P. bermudensis belongs to the Alphaproteobacteria, a predominant and widely distributed bacterial class in the oceans [12, 13]. P. bermudensis is found under diverse environmental conditions and thus needs efficient adaptation strategies. P. bermudensis HTCC2601 has genes encoding for the production of the major highly polar DMSP and is also a DMSOP producer [10, 11]. To the best of our knowledge, the physiological regulation of the zwitterionic metabolites in P. bermudensis and most other marine bacteriais still unexplored.“

We motivate the use of the alga line 67ff:

Since P. bermudensis is found associated to algae, we decided to investigate here the cellular content of zwitterionic metabolites under stress and upon interaction with the microalgae, Tetraselmis striata [15, 16].

We hope that this illustrates the motivation of our study that is aimed at the fundamental principles of bacterial physiology.

Some questions/suggestions are listed below:

Line 14 “organic metabolites released by algae” could the authors specify which algae they refer? 

We added the name of algae after “released by the alga Tetraselmis striata

Line 20 “for bacterial survival in challenging environments” which environments? Please be clear

We clarified “under variable environmental conditions”.

We investigated the effect of several stress conditions including different salinities, temperatures, and the exposure to organic metabolites released by associated algae (Tetraselmis striata) on the halophilic heterotrophic bacterium Pelagibaca bermudensis.

Line 34 “cell organelles against various forms of stress” the sentence appears confused. Why do the authors cite organelles if the topic is the bacterium?

Thank you. We simplified the sentence:

“Osmolytes are naturally occurring organic compounds that protect cells against various forms of stress [5-7].”

Line 60-61 the authors assessed that the Pelagibacter bermudensis “has genes encoding for the production of the major highly polar DMSP and is also a DMSOP producer” why no genetic analysis was performed in this study?

We use this reference to indicate that this bacterium produces DMSP and other possible related osmolytes. Our study focuses on the chemistry of the bacterium, it is orienting clearly towards the analysis of the metabolome. An additional focus on genetics would double the effort of others and extend the manuscript beyond the expertise of the chemical analysis.

the entire manuscript appears out off-topic with respect to the title “ Orchestrated Regulation of Intracellular Zwitterionic Metabolites in Stress Adaptation of the Halophilic Heterotrophic Bacterium Pelagibacter bermudensis” No regulation analysis has been performed but only a response analysis to various conditions. 

We agree that the Regulation is not the core topic but the response of the metabolome. We thus changed the title into “Orchestrated Response of Intracellular Zwitterionic Metabolites in Stress Adaptation of the Halophilic Heterotrophic Bacterium Pelagibacta bermudensis”. We also re-phrased the text wherever “Regulation” was mentioned

Line 203 is not a temperature range the authors examine only three temperatures why? If the objective was to investigate the physiology of the P. bermudensis is strange examine a so far temperature “38°C” from the environmental and optimal temperature. Please could the authors clarify this point?

We now specify:

The examined temperature regime of 18 ˚C, 28 °C, and 38 ˚C represents a selection of conditions encountered by the mesophilic halophilic bacterium P. bermudensis in the natural environment [13]. P. bermudensis is widely distributed in the oceans and found under diverse environmental conditions and thus needs efficient adaptation strategies.  The bacterium grows and survives in a temperature range between 10 °C and 40 °C with an optimum at 28 °C [13]. I accordance, growth rates in our experiments were high at 28 °C and 38 °C and lower at 18 °C.

Line 272-274 the period is very confusing please rewrite

Re-phrased into four sentences

Round 2

Reviewer 3 Report

The authors have improved the quality of the manuscript. However, the style should be revised to be more readable  some other minor suggestions/questions are reported below:

Line 177 The authors should indicate the figure in brackets to better view the results 

Line 187-188 the sentence appears confused please rewrite

Line 189 the conclusions appear inconsistent please clarify

Line 214- 215 “with a 214 optimum at 28 °C” the reference cited by the authors report different optimal (optimally at 30–33 °C) temperature please revise the reference or maybe is new data? 

Author Response

We thank the reviewer for the comments.

Our adjustement follow all the comments.

The style is revised throughout the manuscript and several sentences were simplified.

Line 177 (a.u.) is put in bracketts.

Line 187-190 has been re-written and simplified

Line 214 the temperature optimum is corrected according to the reference.